# Benzodioxane–Benzamides as FtsZ Inhibitors: Effects of Linker’s Functionalization on Gram-Positive Antimicrobial Activity

**DOI:** 10.3390/antibiotics12121712

**Published:** 2023-12-08

**Authors:** Lorenzo Suigo, William Margolin, Eugenia Ulzurrun, Martina Hrast Rambaher, Carlo Zanotto, Victor Sebastián-Pérez, Nuria E. Campillo, Valentina Straniero, Ermanno Valoti

**Affiliations:** 1Dipartimento di Scienze Farmaceutiche, Università degli Studi di Milano, Via Luigi Mangiagalli, 25, 20133 Milano, Italy; lorenzo.suigo@unimi.it (L.S.); ermanno.valoti@unimi.it (E.V.); 2Department of Microbiology and Molecular Genetics, McGovern Medical School, University of Texas, Houston, TX 77030, USA; william.margolin@uth.tmc.edu; 3Centro de Investigaciones Biológicas Margarita Salas, Consejo Superior de Investigaciones Científicas (CSIC), Ramiro de Maeztu 9, 28040 Madrid, Spain; mariaeugenia.ulzurrun@cib.csic.es (E.U.); victorsebastianperez@gmail.com (V.S.-P.); nuria.campillo@csic.es (N.E.C.); 4Centro Nacional de Biotecnología, Consejo Superior de Investigaciones Científicas (CSIC), Darwin 3, 28049 Madrid, Spain; 5Faculty of Pharmacy, University of Ljubljana, Aškerčeva Cesta, 7, 1000 Ljubljana, Slovenia; martina.hrast-rambaher@ffa.uni-lj.si; 6Dipartimento di Biotecnologie Mediche e Medicina Traslazionale, Università degli Studi di Milano, Via Vanvitelli, 32, 20129 Milano, Italy; carlo.zanotto@unimi.it; 7Exscientia, The Schrödinger Building, Oxford Science Park, Oxford OX4 4GE, UK; 8Instituto de Ciencias Matemáticas, Consejo Superior de Investigaciones Científicas (CSIC), C. Nicolás Cabrera, 13-15, 28049 Madrid, Italy

**Keywords:** Gram-positive-dependent diseases, antimicrobial resistance, cell division protein FtsZ, benzamide, 1,4-benzodioxane, multidrug-resistant *Staphylococcus aureus*, *Bacillus subtilis*

## Abstract

FtsZ is an essential bacterial protein abundantly studied as a novel and promising target for antimicrobials. FtsZ is highly conserved among bacteria and mycobacteria, and it is crucial for the correct outcome of the cell division process, as it is responsible for the division of the parent bacterial cell into two daughter cells. In recent years, the benzodioxane–benzamide class has emerged as very promising and capable of targeting both Gram-positive and Gram-negative FtsZs. In this study, we explored the effect of including a substituent on the ethylenic linker between the two main moieties on the antimicrobial activity and pharmacokinetic properties. This substitution, in turn, led to the generation of a second stereogenic center, with both *erythro* and *threo* isomers isolated, characterized, and evaluated. With this work, we discovered how the hydroxy group slightly affects the antimicrobial activity, while being an important anchor for the exploitation and development of prodrugs, probes, and further derivatives.

## 1. Introduction

The development of molecules able to interfere with microbial infections was one of the most important achievements of modern medicine, but, nowadays, the dissemination of Antimicrobial Resistance (AMR) is heavily threatening this important success [1]. During 2019 alone, 1.2 million people died as a direct consequence of AMR, making AMR a leading cause of death worldwide, with a mortality rate higher than malaria and HIV [2].

The ESKAPE pathogens include *Enterococcus faecium*, *Staphylococcus aureus*, *Klebsiella pneumoniae*, *Acinetobacter baumannii*, *Pseudomonas aeruginosa,* and *Enterobacter* species, whose initials make the ESKAPE acronym. These are considered particularly worrying due to their increasing resistance towards several antimicrobials [2]. Among them, *S. aureus* deserves special mention, as it is responsible for severe nosocomial infections, significantly increasing the time of hospitalization, costs, and mortality. Recently, *S. aureus* has also become resistant to vancomycin and several other last-resort drugs, worsening the overall situation [3].

As suggested by the World Health Organization, one of the best potential strategies to overcome AMR is the development of molecules able to bind to new molecular targets that have not yet exploited by marketed antibiotics. In this regard, the bacterial cell division cycle is well known as a very promising target: as most bacteria divide through binary fission, the cell division machinery is highly conserved among different species, providing the potential for broad-spectrum agents [4,5]. FtsZ, a tubulin-like protein, is a key factor for organizing the cell division process. It initially polymerizes into a “Z ring” at the center of the cell. This first step is followed by the recruitment of other important cell division proteins that assemble the mature divisome, which, in turn, synthesizes the division septum that splits the original cell into two daughter cells [6,7]. FtsZ and the Z ring are crucial for this process, as inhibition of FtsZ polymerization or dynamics results in failed division, cell filamentation, and lysis. Several structurally distinct small-molecule inhibitors of FtsZ have been studied and developed over the last 15 years. Among them, the benzamide class is the most studied, thanks to its chemical accessibility, relatively low cytotoxicity, and positive results obtained with prototypes, such as 2,6-difluoro-3-nonyloxybenzamide and PC190723 (Figure 1) [8]. In particular, PC190723 strongly inhibits *S. aureus* FtsZ function in cell division, with a Minimal Inhibitory Concentration (MIC) of 1 μg/mL towards both methicillin-sensitive *Staphylococcus aureus* (MSSA) and methicillin-resistant *S. aureus* (MRSA) [8].

Starting from these molecules, we recently first replaced the thiazolopyridine with an unsubstituted 1,4-benzodioxane ring, giving rise to **FZ14** (Figure 2) [9], and then to several differently functionalized 1,4-benzodioxanes [10,11,12,13,14], resulting in the benzodioxane–benzamide FtsZ inhibitors class. The versatility of the 1,4-benzodioxane ring, a well-known scaffold in medicinal chemistry, led to the simple introduction of several structural modifications. Development of a robust and reliable computational model [13] provided the impetus to lengthen the methylenoxy to an ethylenoxy linker (**FZ14** to **FZ88**), which resulted in strongly increasing the antimicrobial activity (Figure 2) [12]. A comprehensive Structure Activity Relationship (SAR) study led to the development of very potent compounds [14], named **FZ95** and **FZ100** (Figure 2), in which the benzamide pharmacophoric structure is linked to a naphthodioxane or a 5,6,7,8-tetrahydronaphthodioxane ring by an ethylenoxy linker.

**FZ95** and **FZ100** are characterized by having very high potencies against both MRSA and MSSA, with MICs of 0.25 μg/mL (0.6 μM) and 0.1 μg/mL (0.25 μM), respectively. They also showed MICs under 0.1 μg/mL vs. *B. subtilis,* and their activity is accompanied by low to no cytotoxicity towards human MRC-5 cells [14]. In addition, we recently demonstrated how both of these compounds are able to interact with *E. coli* FtsZ and to interfere with its essential properties, therefore exerting antimicrobial activity on efflux-pump-defective *E. coli* strains, such as N43 (Δ*acrAB*) and Δ*tolC* [15]. These optimal microbiological profiles make these compounds very intriguing as potential antibiotics. Nevertheless, **FZ95** and **FZ100** have high lipophilicity, which will very likely translate to poor water solubility, poor oral bioavailability, a tendency to bioaccumulate, and other undesired pharmacokinetic properties. For both PC190723 and TXA707, which are characterized by similar problems, prodrugs have been designed by exploiting the benzamide group and converting it into a labile imide moiety [16]. Moreover, Stokes and collaborators, while working on the development of bromo-oxazole benzamides as FtsZ inhibitors [17,18], demonstrated how the introduction of hydroxy-methyl in the methylene bridge of the molecule resulted in an almost 10-fold increase in solubility without diminishing the antimicrobial activity. Moreover, the further derivatization with a liable succinic ester moiety redoubled the solubility value [17,18]. Starting from these findings, we continued the development of naphthodioxane–benzamides as FtsZ inhibitors by focusing on improving the physical–chemical properties. As a result, we recently designed an innovative set of derivatives characterized by an additional -OH on the linker between the two main moieties. This modification introduces a second stereogenic center, prompting us to isolate and individually evaluate the *erythro* and *threo* isomers. As reported in our recent work focused on *E. coli* as a target, **FZ116** (Figure 3) resulted in very strong in vitro inhibition of *E. coli* FtsZ and inhibition of cell division in efflux-pump-defective *E. coli* [15]. Stokes and co-workers also observed how the introduction of a methyl group in the same position resulted in the enhancement of the antimicrobial activity vs. *S. aureus* without positively affecting the solubility [17]. These results suggest how both hydrophilic and lipophilic substituents could be tolerated.

Starting from these considerations, in this work, we focused our attention on the effects of introducing different linker substituents on the Gram-positive antimicrobial activity of these compounds. We firstly determined how introducing a methyl group on the structures of **FZ14**, **FZ88**, and **FZ100** influenced the antimicrobial activity. As the introduction of a methyl group resulted in a second stereogenic center, we thus characterized each *erythro* and *threo* isomer (derivatives **FZ104**, **FZ105**, **FZ98**, **FZ97**, **FZ118**, and **FZ119** in Figure 3). In addition, we evaluated the effects of each novel compound, as well as those of **FZ116**, **FZ117**, **FZ112**, and **FZ113** (Figure 3), which were previously described as bearing the -OH group as a substituent, as antimicrobials against Gram-positive *S. aureus* and *B. subtilis*.

## 2. Results

### 2.1. Chemistry

Figure 1 reports the synthetic pathway for achieving compounds **FZ104** and **FZ105**, which started from methyl ketone **1**, an intermediate developed by our research team and whose preparation was previously described and applied [19]. The methyl ketone **1** underwent reduction (**2**) and further mesylation, thus obtaining compound **3,** which was substituted with the pharmacophoric 2,6-difluoro-3-hydroxybenzamide (*in house* prepared [9]). The separation of the *erythro* and the *threo* isomers was achieved using flash chromatography on silica gel, although further synthetic work was required for the identification of the correct isomer. We therefore synthesized the pure (*S*,*R*) *erythro* enantiomer, as described in Figure 2.

We started with the commercially available *D*-Mannitol diacetonide, which was converted into the aldehyde **4** through the oxidative C-C bond cleavage using NaIO_4_. This intermediate was then reduced (**5**), and the hydroxylic function was benzylated (**6**). The two isomers were then isolated through flash chromatography, and the (*R*,*S*) stereoisomer was identified by comparing its ^1^H-NMR spectrum in CDCl_3_ and its optical properties with the limited data from the literature [20,21]. The synthesis then continued with the (*R*,*S*) isomer through the hydrolysis of the acetonide, accomplishing intermediate **7** (a comparison of ^1^H-NMR chemical shifts with a different literature reference [22] allowed the further confirmation of its absolute configuration) and mesylation of both the hydroxy functions (**8**). These were rapidly substituted with catechol, achieving the 1,4-benzodioxane ring with a complete inversion of the configuration of the benzodioxane stereocenter (**9**). The final debenzylation (**10**), mesylation (**11**), and substitution with the pharmacophoric 2,6-difluoro-3-hydroxybenzamide allowed for the obtainment of enantiopure (*S*,*R*)-**FZ105**.

To obtain compounds **FZ98** and **FZ97** (Figure 3), the synthesis started from derivative **1,** which underwent the Wittig reaction (**12**) and anti-Markovnikov water addition (**13**). Following the strategy described for **FZ104** and **FZ105,** the hydroxy function was mesylated (**14**) and substituted with the pharmacophoric benzamide. At this point, the two isomers were isolated on preparative HPLC, and their configuration was easily assessed through comparison of ^1^H-NMR spectra and chromatographic data, therefore defining **FZ98** as *erythro* and **FZ97** as *threo*.

The syntheses of derivatives **FZ118** and **FZ119** followed a complete novel synthetic strategy, as reported in Figure 4. After having applied an *in house* previously described synthetic strategy [23] and having achieved the 5,6,7,8-tetrahydronaphthalen-2,3-diol, this was reacted with methyl 3,4-dibromo-2-methylbutanoate (**17**), which was obtained through deconjugative transposition (**15**), bromination (**16**), and esterification of the commercially available tiglic acid. The nature of the derivative obtained allowed for the separation of the two isomers using flash chromatography and the definition of the *erythro* from the *threo* isomer through comparison of ^1^H-NMR spectra. The syntheses proceeded individually for the two isomers through reduction (**19**, **20**), mesylation (**21**, **22**), and condensation with the pharmacophoric 2,6-difluorobenzamide, therefore achieving compounds **FZ118** and **FZ119**.

### 2.2. Antimicrobial Activity on Staphylococcus aureus

As described in our previous work [12,13,14], we evaluated all the compounds mentioned above on different *S. aureus* strains. We tested on both methicillin-sensitive *S. aureus* (MSSA, ATCC29213) and two clinical isolates of *S. aureus* showing multi-drug resistance (MDRSA). MDRSA 12.1 is resistant towards cefoxitin, gentamicin, kanamycin, rifampicin, streptomycin, sulfamethoxazole, and tetracycline, while MDRSA 11.7 shows resistance to cefoxitin, ciprofloxacin, clindamycin, erythromycin, quinupristin-dalfopristin association, tetracycline, tiamulin, and trimethoprim. We analyzed the MICs and, if the antibacterial activity was promising, we also determined the percentage of cytotoxicity using the MTT assay on human MRC-5 cells and reported it as TD90. The results of antibacterial properties of all of the synthesized derivatives are shown in Table 1, and they are compared with their reference compounds **FZ14**, **FZ88**, and **FZ100**.

As observed, neither the introduction of the methyl nor of the hydroxy led to a further strong increase in the antimicrobial activity vs. *S. aureus.* Nonetheless, **FZ116**, the *erythro* hydroxy derivative of **FZ100**, and **FZ118** and **FZ119, the** methyl derivatives of **FZ100**, retain promising activity versus all of the bacterial strains and a negligible degree of cytotoxicity.

### 2.3. Antimicrobial Activity on Bacillus subtilis and Effects of FZ116 and FZ117 on Its Inhibition

We further evaluated the most promising compounds on *B. subtilis*, a model Gram-positive species used previously in evaluating other benzamides [24], including by our research group [14]. The *B. subtilis* strain WM5126 expresses a xylose-inducible GFP-tagged ZapA protein. As ZapA interacts directly with FtsZ, GFP-ZapA serves as a proxy for the Z ring without significantly altering the properties of the FtsZ target itself. We first tested the effects of the compounds on colony viability on solid medium (Figure 4). Although none of the compounds were as potent as **FZ100**, **FZ116** and **FZ119** killed *B. subtilis* at the lowest doses, with **FZ117** and **FZ118** being less effective (Figure 4A). Among the less potent compounds, **FZ104** was the best, as it was able to kill at 6 µg/mL (~20 µM), whereas **FZ98**, **FZ112,** and **FZ113** still permitted growth at this concentration (Figure 4B). MIC values from microdilution experiments, shown in the right column in Figure 4, largely mirrored the spot viability data. The effects of the compounds on *B. subtilis* also generally tracked with their effects on *S. aureus*, with **FZ100** being the most potent, followed by **FZ116**, **FZ119,** and **FZ118**, and with **FZ112**, **FZ113**, and **FZ98** having the lowest antimicrobial activities.

To test if the differential effects of **FZ116** and **FZ117** enantiomers could be detected as FtsZ assembly changes in *B. subtilis* cells, we visualized GFP-ZapA patterns after growth in inhibitory concentrations of **FZ116** and **FZ117** (0.3 µg/mL or 0.8 µM). Cells grown with no added compound grew as typical single bacilli and short chains, with well- defined Z rings clearly visible (Figure 5, left panel, single yellow arrows). In contrast, FtsZ rings were largely disrupted in cells with **FZ116**, with most of the fluorescence localized to speckles (middle panel, forked arrows) with only rare intact Z rings (yellow arrows). The lower potency of **FZ117** was manifested by a mixture of FtsZ speckles and Z rings, although, as with **FZ116**-treated cells, most of the cells were filamentous because of division defects. This conversion of FtsZ rings into a speckle pattern was observed previously, both with **FZ100** [14] and with the original benzamide derivatives [24].

### 2.4. Computational Studies

#### 2.4.1. Physicochemical and Drug-like Profile Calculations

A wide variety of parameters able to cover all of the possible concerns of these interesting derivatives were chosen and predicted. We also considered and calculated an important number of the most relevant molecular and physicochemical properties, as well as some important pharmacokinetics predictions, as summarized in Table 2 and Table 3.

The majority of the novel proposed FtsZ inhibitors showed appropriate physicochemical qualities and a desirable drug-like profile, meeting the rule of five with no outliers. Moreover, none of the compounds showed issues related to permeability or potential hERG liability. Therefore, considering these promising properties, all of these compounds represent a good starting point for further optimization efforts.

#### 2.4.2. Docking Studies

To explain the binding mode and the biological activity of these compounds, a computational study was performed comprising a docking study followed by a more in-depth study of Induced Fit Docking (IFD). The first step was the validation of the protocol. For this purpose, the previously described TXA707 compound was redocked into the binding site of the FtsZ crystal structure. The sole pose showed a docking score of −12.831 Kcal/mol and an almost complete superposition of the ligand TXA707 in its crystallographic or docked form (Figure 6).

After validating the docking protocol, our precursor **FZ100** was then docked, and it was confirmed that the 2,6-difluorobenzamide moiety could establish the three key hydrogen bonds (HB) with the *S. aureus* FtsZ protein: the primary amide acted as a Hydrogen Bond Donor (HBD) with Val207 and Asn263, while the carbonyl function acted as a Hydrogen Bond Acceptor (HBA) with Leu209 (Table 4). Having validated the docking and IFD protocols, an IFD for each new inhibitor was performed. The results are summarized in Appendix A, which outlines promising results for all of the poses of this new family of inhibitors, thus confirming the maintenance of the three key hydrogen bonds.

### 2.5. Linker Effect about the Accomodation into the Binding Pocket

To understand the importance and the role of OH and CH_3_ substituents, especially on the influence of the molecule accommodation into the binding site, we analyzed all of the possible aminoacidic residues in the radius of 3 Å around the poses able to form the three key HBs. This allowed us to identify two different clusters, depending on the aminoacidic environment and, consequently, on the potential additional interactions (Table 5). The former group is characterized by the clear and definite binding interaction with the residue Thr309 (Figure 7A) via a HB, and the latter by the absence of residues around the substituent (Figure 7B). In addition, two peculiar patterns should be defined between the *erythro* and *threo* isomers bearing the OH substituent, although the presence of a weak HB interaction with Thr309 was prevalent in the *threo*-OH group.

Finally, we superposed **FZ100** with the two new inhibitors clusters, therefore allowing us to identify a novel and peculiar behavior involving the binding mode of the benzodioxane moiety. It is interesting to highlight that all poses, except for **FZ104**, show a 180º rotation of the benzodioxane moiety into the hydrophobic binding subpocket driven by the ethylenoxy linker (Appendix A and Appendix A). This highlights that the binding and conformation are driven by the three key HB interactions in the benzamide pocket and the lipophilic tail in the hydrophobic subpocket. Therefore, the linker acts as a spacer to accommodate the key pharmacophoric features in the molecule.

This unexpected harboring is easily visible in Figure 7, where the red circles highlight the twisting of the linker, thus allowing the 180° rotation of the tetrahydronaphthodioxane ring plane around its axis of symmetry.

## 3. Discussion

The evaluation of antimicrobial assays on both *S. aureus* and *B. subtilis*, together with the predicted properties, the docking poses, and the scores, led us to several interesting conclusions.

First, the development of the differentially substituted **FZ116**, **FZ117**, **FZ118** and **FZ119**, the tetrahydro naphthodioxane compounds, allowed us to obtain promising derivatives with antimicrobial activities against both *S. aureus* and *B. subtilis*. This consideration, together with the recent results with efflux-deficient *E. coli* strains [15], strongly pave the way to FtsZ inhibitors able to possess a wide spectrum of action not restricted to exclusively Gram-positive or Gram-negative strains.

Different hypotheses should be tested depending on the substituent inserted on the ethylenoxy linker, because methyl or hydroxy substituents in turn led to different outcomes, both in terms of antimicrobial potencies and computational properties.

Alternately, the introduction of a methyl group does not result in any possible additional interaction with aminoacidic residues of the protein to further strengthen the harboring into the binding site; consequently, this did not lead to an increase in the antimicrobial activity. The main effect of its insertion is in terms of spatial volume, because this functional group is more hindered than the hydroxy function and spatially spherical, without any possibility to be oriented by the aminoacidic environment around it. As a result, no differences between *erythro* and *threo* isomers in terms of antimicrobial activities could be detected.

A completely different outcome was found for the OH derivatives, because several possible aminoacidic residues, especially Threonine 309, are present within the binding site and could allow for the establishment of additional hydrogen bonds. Moreover, the hydroxy group is less hindered and, more importantly, it is angular and it could be differently oriented, as it could be driven by surrounding residues.

The results achieved in this work are perfectly in line with the computational model developed so far [12,13,14]. The model suggested to us how the FtsZ binding subpocket, responsible for harboring the benzodioxane or tetrahydronaphthodioxane, is characterized by hydrophobic and non-aromatic residues and how the linker between the two aromatic scaffolds is deputed to the best accommodation into this hydrophobic cavity.

Here, we further demonstrated how the linker is not involved in specific interactions and how it is able to freely orientate itself to achieve the best accommodation of the naphthodioxane moiety. Moreover, considering the symmetric and linear nature of the tetrahydronaphthodioxane, the degree of freedom is even higher, with the possibility of performing a 180° rotation around the axis of symmetry to maximize the interaction within the protein binding pocket. Finally, focusing on the best computational poses of **FZ116** and **FZ117**, in both of the isomers, it seems that the 180° rotation induces the OH group orientation in order to have the polar substituent out of the binding pocket and available for further derivatization, such as the preparation of prodrugs or probes.

## 4. Materials and Methods

### 4.1. Chemistry

The totality of the reagents and the solvents were purchased from commercial sources (Merck KGaA (Darmstadt, Germany), Fluorochem (Hadfield, United Kingdom), and Carlo Erba (Cornaredo, Milan, Italy)), and they were used without further purification or distillation.

Silica gel matrix was used in both TLC (thin-layer chromatography on aluminum foils having a fluorescent indicator 254 nm) and in flash chromatography (particle size 230–400 mesh, Carlo Erba) on Puriflash XS 420 (Sepachrom Srl, Rho (MI), Italy). The visualization was performed with UV light at 254 nm or at 280 nm (λ).

A Varian (Palo Alto, CA, USA) Mercury 300 NMR spectrometer/Oxford Narrow Bore superconducting magnet operating at 300 MHz was used for all ^1^H-NMR spectra. ^13^C-NMR spectra were acquired at 75 MHz. The chemical shifts are reported in ppm (δ), relative to the residual solvent as an internal standard. The following abbreviations refer to signal multiplicity: s = singlet, d = doublet, dd = doublet of doublets, ddd = doublet of doublet of doublets, dq = doublet of quadruplets, quint = quintet, m = multiplet, bs = broad singlet.

The final products (**FZ104**, **FZ105**, **FZ97**, **FZ98**, **FZ118,** and **FZ119)** were analyzed through reverse-phase HPLC using a Waters XBridge C-18 column (5 µm, 4.6 mm × 150 mm) on an Elite LaChrom HPLC system (Hitachi, San Jose, CA; USA) equipped with a DAD (diode array detector). HPLC methods used two different solvents: A: H_2_O with 0.10% TFA, and B: acetonitrile with 0.10% TFA. Method A: isocratic elution 60% A and 40% B with 30 min run time and a flow rate of 1 mL/min. Method B: isocratic elution 70% A and 30% B with 40 min run time and a flow rate of 1.2 mL/min. Method C: gradient elution from 90% A to 10% A in 25 min with 35 min run time and a flow rate of 1 mL/min. The purities of the final products was quantified at specific λ values, depending on the compound, and all were >95%. The relative retention times are reported in each experimental section. The melting points were determined through DSC analysis using a DSC 1020 apparatus (TA Instruments, New Castle, DE, USA).

The ^1^H- and ^13^C-NMR spectra of all of the final compounds, together with their HPLC profiles, are included in the Appendix A.

#### Synthesis

**1-hydroxyethyl-1,4-benzodioxane (2):** NaBH_4_ (0.21 g, 5.61 mmol) was added at 0 °C to a solution of 2-acetyl-1,4-benzodioxane (2.00 g, 11.22 mmol) in methanol (20 mL). The reaction was kept stirring at 0 °C for 30 min, and then the solvent was concentrated under vacuum to yield 2.02 g (quantitative yield) of **2** as a brown oil. **^1^H-NMR (300 MHz, CDCl_3_):** δ 7.06–6.73 (m, 4H), 4.32 (ddd, *J* = 12.8, 11.3, 2.0 Hz, 1H), 4.17–3.92 (m, 3H), 1.34 (d, *J* = 6.5 Hz, 3H) ppm.

**1-mesiloxyethyl-1,4-benzodioxane (3):** Methanesulfonyl chloride (1.54 g, 13.4 mmol) was added dropwise to a solution of 1-hydroxyethyl-1,4-benzodioxane **2** (2.02 g, 11.2 mmol) and TEA (1.48 g, 14.6 mmol) in DCM (40 mL) at 0 °C. The mixture was stirred at room temperature for 3 h, diluted with DCM, washed first with 10% aqueous NaHCO_3_, second with 10% aqueous HCl, and finally with brine, dried over Na_2_SO_4_, filtered, and concentrated under vacuum to yield 2.65 g (92%) of **3** as a brown oil. **^1^H-NMR (300 MHz, CDCl_3_):** δ 6.95–6.77 (m, 4H), 4.96 (m, 1H), 4.50–4.00 (m, 3H), 3.07 (s, 3H), 1.56 (d, *J* = 6.6 Hz, 3H) ppm.

***Erythro* (FZ104) and *threo* (FZ105) 3-[1-(1,4-benzodioxan-2yl)ethyl-1-oxy]-2,6-difluorobenzamide:** Operating under a nitrogen atmosphere, potassium carbonate (0.84 g, 6.1 mmol) was added to a solution of 2,6-difluoro-3-hydroxybenzamide (1.00 g, 5.8 mmol) in dry DMF (5 mL). After stirring at room temperature for 30 min, a solution of 1-mesiloxyethyl-1,4-benzodioxane **3** (1.49 g, 5.8 mmol) in DMF (5 mL) was added. The reaction mixture was stirred at 80 °C for 16 h, concentrated under vacuum, diluted with ethyl acetate (30 mL), washed with brine (5 × 20 mL), dried over Na_2_SO_4_, filtered, and concentrated, to give 1.53 g of 3-[1-(1,4-benzodioxan-2yl)ethyl-1-oxy]-2,6-difluorobenzamide as a mixture of *erythro* and *threo* isomers. The crude was purified through flash chromatography on silica gel by eluting with 6/4 Cyclohexane/Ethyl Acetate, giving 0.28 g of pure *erythro* isomer and 0.12 of pure *threo* isomer (20% cumulative yield).

***Erythro* FZ104**: **^1^H-NMR (300 MHz, DMSO-d_6_):** δ 8.10 (bs, 1H), 7.82 (bs, 1H), 7.32 (dt, *J* = 9.2, 5,3 Hz, 1H), 7.05 (dt, *J* = 9.2, 1.9 Hz, 1H), 6.92–6.75 (m, 4H), 4.66 (dq, *J* = 6.3, 5.1 Hz, 1H) 4.47 (dd, *J* = 11.5, 2.2 Hz, 1H), 4.32 (ddd, *J* = 7.3, 5.1, 2.2 Hz, 1H), 4.10 (dd, *J* = 11.5, 7.3 Hz, 1H), 1.36 (d, *J* = 6.3 Hz, 3H) ppm. **^13^C-NMR (75 MHz, DMSO-d_6_):** δ 161.7, 152.8 (dd, *J* = 242.2, 6.8 Hz), 149.4 (dd, *J* = 248.6, 8.6 Hz), 143.5, 143.3, 142.0 (dd, *J* = 11.0, 3.3 Hz), 122.0, 121.8, 118.7 (dd, *J* = 9.4, 2.2 Hz), 117.5, 117.4, 117.2 (dd, *J* = 24.8, 20.9 Hz), 111.6 (dd, *J* = 23.0, 4.0 Hz) 75.3, 75.1, 64.4, 16.0 ppm. Tr (HPLC): 7.44 min (Method A).

***Threo* FZ105: ^1^H-NMR (300 MHz, DMSO-d_6_):** δ 8.11 (bs, 1H), 7.83 (bs, 1H), 7.27 (dt, *J* = 9.1, 5,4 Hz, 1H), 7.03 (dt, *J* = 9.1, 1.6 Hz, 1H), 6.91–6.77 (m, 4H), 4.65 (dq, *J* = 6.3, 5.2 Hz, 1H) 4.39 (dd, *J* = 11.3, 2.1 Hz, 1H), 4.33 (ddd, *J* = 7.4, 5.2, 2.1 Hz, 1H), 4.10 (dd, *J* = 11.3, 7.4 Hz, 1H), 1.35 (d, *J* = 6.3 Hz, 3H) ppm. **^13^C-NMR (75 MHz, DMSO-d_6_**): δ 161.7, 152.8 (dd, *J* = 242.1, 6.9 Hz), 149.5 (dd, *J* = 248.6, 8.6 Hz), 143.5, 143.3, 142.3 (dd, *J* = 11.0, 3.4 Hz), 121.9, 121.7, 119.1 (dd, *J* = 9.4, 2.1 Hz), 117.6, 17.3, 117.2 (dd, *J* = 24.9, 20.9 Hz), 111.6 (dd, *J* = 23.0, 3.9 Hz) 75.8, 75.4, 64.9, 15.8 ppm. Tr (HPLC): 6.01 min (Method A).

**(*R*)-2,2-dimethyl-1,3-dioxolane-4-carbaldehyde (4):** NaIO_4_ (6.11 g, 28.59 mmol) was added in portion to a solution of *D*-mannitol diacetonide (5.00 g, 19.06 mmol) and 10% aqueous NaHCO_3_ (3 mL) in DCM (50 mL). The reaction was left stirring for 5 h, and then the small amount of water was removed through treatment of the reaction mixture over Na_2_SO_4_, filtration, and concentration under mild vacuum, yielding 4.96 g (quantitative yield) of compound **4** as a yellow oil. **^1^H-NMR (300 MHz, CDCl_3_):** δ 9.71 (d, *J* = 1.8 Hz, 1H), 4.42–4.34 (m, 1H), 4.22–4.06 (m, 3H), 1.48 (s, 3H), 1.41 (s, 3H) ppm. [α]*_D_*^25^ (c 1.0, CHCl_3_) = + 38. (Lit +40 (c 1.0, CHCl_3_) [25]).

**(1*R,*4*R)* and (1*S,*4*R*)-1-(2,2-dimethyl-1,3-dioxolan-4-yl)ethanol (5):** Operating under a nitrogen atmosphere, a solution of methylmagnesium bromide 3 M in THF (19 mL) was added dropwise at 0 °C to a solution of (*R*)-2,2-dimethyl-1,3-dioxolane-4-carbaldehyde **4** (4.96 g, 38.11 mmol) in dry THF (100 mL). After stirring at RT for 1.5 h, the reaction was quenched with H_2_O/Et_2_O and extracted with Et_2_O twice. The organic layer was dried over Na_2_SO_4_, filtered, and concentrated under vacuum to yield 3.6 g (64%) of a yellow oil consisting of compound **5** as a mixture of isomers. **^1^H-NMR (300 MHz, CDCl_3_):** δ 4.07–3.85 (m, 3H), 3.70 (m, 1H), 1.43 (s, 3H), 1.37 (s, 3H), 1.16 (d, *J* = 6.3 Hz, 3H), 1.15 (d, *J* = 6.3 Hz, 3H) ppm.

**(4*R,*1′*S*)-4-(1-(benzyloxy)ethyl)-2,2-dimethyl-1,3-dioxolane and (4*R,1′R*)-4-(1-(benzyloxy)ethyl)-2,2-dimethyl-1,3-dioxolane (6):** Operating under a nitrogen atmosphere, a solution of 1-((*R*)-2,2-dimethyl-1,3-dioxolan-4-yl)ethanol **5** (1.0 g, 6.84 mmol) in dry THF (10 mL) was added dropwise to a suspension of NaH 60% (0.41 g, 10.26 mmol) at 0 °C. After stirring for 30 min, benzyl bromide (1.44 mL, 7.87 mmol) was added dropwise at 0 °C, and the reaction was left stirring for 15 min before warming it to RT. After 2 h, the reaction mixture was cooled down to 0 °C, quenched with H_2_O and ethyl acetate, and the aqueous phase was extracted twice with ethyl acetate. The organic phase was dried over Na_2_SO_4_, filtered, and concentrated under vacuum to yield 1.7 g of crude **6** that was purified through flash chromatography on silica gel. Elution with 9/1 cyclohexane/ethyl acetate gave 0.8 g of **(4*R,*1′*S*)-4-(1-(benzyloxy)ethyl)-2,2-dimethyl-1,3-dioxolane 6-*erythro*** (50%) and 0.4 g of **(4*R,*1′*R*)-4-(1-(benzyloxy)ethyl)-2,2-dimethyl-1,3-dioxolane 6-*threo*** (25%).

**(4*R,*1′*S*)-4-(1-(benzyloxy)ethyl)-2,2-dimethyl-1,3-dioxolane (6-*erythro*): ^1^H-NMR (300 MHz, CDCl_3_):** δ 7.40–7.27 (m, 5H), 4.64 (d, *J* = 11.7 Hz, 1H), 4.51 (d, *J* = 11.7 Hz, 1H), 4.07 (dt, *J* = 6.2, 5.5 Hz, 1H), 4.00 (dd, *J* = 7.6, 6.2 Hz, 1H), 3.87 (dd, *J* = 7.6, 5.5 Hz, 1H), 3.54 (quint, *J* = 6.2 Hz 1H), 1.41 (s, 3H), 1.36 (s, 3H), 1.25 (d, *J* = 6.2 Hz, 3H) ppm. [α]*_D_*^25^ = + 42.8 (c 1.23, EtOH). (Lit +43.4 (c 4.6, EtOH) [20]).

**(4*R,*1′*R*)-4-(1-(benzyloxy)ethyl)-2,2-dimethyl-1,3-dioxolane (6-*threo*): ^1^H-NMR (300 MHz, CDCl_3_):** δ 7.40–7.27 (m, 5H), 4.68 (d, *J* = 12.2 Hz, 1H), 4.63 (d, *J* = 12.2 Hz, 1H), 4.17 (dt, *J* = 6.4, 7.0 Hz, 1H), 3.99 (dd, *J* = 8.3, 6.4 Hz, 1H), 3.72 (dd, *J* = 8.3, 7.0 Hz, 1H), 3.61 (quint, *J* = 6.4 Hz 1H), 1.43 (s, 3H), 1.37 (s, 3H), 1.13 (d, *J* = 6.4 Hz, 3H) ppm. [α]*_D_*^25^= + 7.5 (c 0.95, EtOH).

**(2*R*,3*S*)-3-benzyloxybutane-1,2-diol (7-*erythro*):** A solution of H_2_SO_4_ 2.0 M (4.5 mL) was added to a solution of *(4R*,*1′S)*-4-(1-(benzyloxy)ethyl)-2,2-dimethyl-1,3-dioxolane **6** (0.7 g, 2.96 mmol) in dioxane (7 mL). After stirring at 110 °C for 2 h, the reaction mixture was concentrated under vacuum, phosphate buffer was added, and the aqueous layer was made basic and extracted with ethyl acetate twice. The organic phase was dried over Na_2_SO_4_, filtered, and concentrated under vacuum to yield 0.45 g (78%) of compound **7-*erythro*** as a brown oil. **^1^H-NMR (300 MHz, CDCl_3_):** δ 7.45–7.28 (m, 5H), 4.65 (d, *J* = 11.6 Hz, 1H), 4.46 (d, *J* = 11.6 Hz, 1H), 3.85–3.56 (m, 4H), 1.23 (d, *J* = 6.2 Hz, 3H) ppm. [α]*_D_*^25^ = +32.8 (c 0.81, CH_3_Cl).

**(2*R*,3*S*)-3-benzyloxy-1,2-dimethanesulfonyloxybutane (8-*erythro*):** Methanesulfonyl chloride (0.52 mL, 6.72 mmol) was added dropwise to a solution of (2*R*,3*S*)-3-(benzyloxy)butane-1,2-diol **7-*erythro*** (0.44 g, 2.24 mmol) and TEA (0.94 g, 6.72 mmol) in DCM (5 mL) at 0 °C. The mixture was stirred at room temperature for 3 h, diluted with DCM, washed with brine, dried over Na_2_SO_4_, filtered, and concentrated under vacuum to yield 0.65 g (92%) of compound **8-*erythro*** as a brown oil. **^1^H-NMR (300 MHz, CDCl_3_):** δ 7.45–7.28 (m, 5H), 4.65 (d, *J* = 11.6 Hz, 1H), 4.46 (d, *J* = 11.6 Hz, 1H), 3.85–3.56 (m, 4H), 1.23 (d, *J* = 6.2 Hz, 3H) ppm. [α]*_D_*^25^ =+20.5 (c 0.95, CH_3_Cl).

**(2*S,*1′*S*)-2-(1′-(benzyloxy)ethyl)-1,4-benzodioxane (9-*erythro*):** Potassium carbonate (0.59 g, 4.26 mmol) was added to a solution of catechol (0.16 g, 1.42 mmol) in acetone (5 mL). After stirring at RT for 30 min, a solution of (2*R*,3*S*)-3-(benzyloxy) -1,2-dimethanesulfonylbutane **8-*erythro*** (0.5 g, 1.42 mmol) in acetone (2 mL) was added dropwise, and the reaction mixture was heated to 60 °C for 18 h. The reaction mixture was concentrated under vacuum, diluted with ethyl acetate, and washed with H_2_O. The organic layer was dried over Na_2_SO_4_, filtered, and concentrated under vacuum to yield 0.4 g, and the obtained crude was purified through flash chromatography on silica gel. Elution with 95/5 cyclohexane/ethyl acetate gave 0.14 g (40%) of compound **9-*erythro*** as a yellow oil. **^1^H-NMR (300 MHz, CDCl_3_):** δ 7.40–7.27 (m, 5H), 6.95–6.80 (m, 4H), 4.69 (d, *J* = 11.9 Hz, 1H), 4.58 (d, *J* = 11.9 Hz, 1H), 4.27 (dd, *J* = 10.7, 1.8 Hz, 1H), 4.18 (ddd, *J* = 7.7, 4.7, 1.7 Hz, 1H), 4.11 (dd, *J* = 10.6, 7.7 Hz, 1H), 3.84 (dq, *J* = 6.5, 4.7 Hz, 1H), 1.32 (d, *J* = 6.5 Hz, 1H). [α]*_D_*^25^ = +30.9 (c 0.54, CH_3_Cl).

**(2*S,*1′*S*)-2-(1′-hydroxyethyl)-1,4-benzodioxane (10-*erythro*):** Operating under a hydrogen atmosphere, Pd/C 5 wt.% loading (0.014 g) was added to a solution of (*2S,1′S*)-1′-(benzyloxy)ethyl)-1,4-benzodioxane **9-*erythro*** (0.14 g, 0.52 mmol) in methanol (20 mL). After stirring for 4 h, the reaction mixture was filtered through a pad of celite, and the solvent was concentrated under vacuum to yield 0.094 g (quantitative yield) of compound **10-*erythro*** as a yellow oil. **^1^H-NMR (300 MHz, CDCl_3_):** δ 7.02–6.72 (m, 4H), 4.28 (dd, *J* = 11.2, 1.9 Hz, 1H), 4.07 (dd, *J* = 11.2, 7.8 Hz, 1H), 4.00–3.93 (m, 2H), 1.34 (d, *J* = 6.3 Hz, 1H) ppm. [α]*_D_*^25^ = +43.0 (c 0.81, CH_3_Cl).

**(2*S,*1′*S*)-2-(1′-methansulfonyloxyethyl)-1,4-bendioxane (11-*erythro*):** Methanesulfonyl chloride (57 μL, 0.74 mmol) was added dropwise to a solution of (*2S,1′S*)-2,3-dihydrobenzodioxan-2-yl)ethanol **10-*erythro*** (89 mg, 0.49 mmol) and TEA (103 μL, 0.74 mmol) in DCM (5 mL) at 0 °C. The mixture was stirred at room temperature for 3 h, diluted with DCM, washed with brine, dried over Na_2_SO_4_, filtered, and concentrated under vacuum to yield 120 mg (quantitative yield) of compound **11-*erythro*** as a brown oil. **^1^H-NMR (300 MHz, CDCl_3_):** δ 6.96–6.78 (m, 4H), 4.93 (quint, *J* = 6.6 Hz, 1H), 4.31- 4.21 (m, 2H), 4.09 (m, 1H), 3.06 (s, 3H), 1.56 (d, *J* = 6.6 Hz, 3H) ppm. [α]*_D_*^25^ = +16.8 (c 0.81, CH_3_Cl).

***Threo* (1*R,*2′*S*)-3-(1-(1′,4′-benzodioxan-2′-yl)ethoxy)-2,6-difluorobenzamide (FZ105 *S′R*):** Operating under a nitrogen atmosphere, potassium carbonate (0.07 g, 0.51 mmol) was added to a solution of 2,6-difluoro-3-hydroxybenzamide (85 mg, 0.49 mmol) in dry DMF (1 mL). After stirring at room temperature for 30 min, a solution of (*2S,1′S*)-2-(1′-methansulfonyloxyethyl)-1,4-bendioxane **11** (120 mg, 0.47 mmol) in DMF (1 mL) was added. The reaction mixture was stirred at 80 °C for 16 h, concentrated under vacuum, diluted with ethyl acetate (30 mL), washed with brine (5 × 20 mL), dried over Na_2_SO_4_, filtered, and concentrated, to give 30 mg (22%) of **FZ105 *SR*** as a waxy residue. [α]*_D_*^25^ = -8.13 (c 0.36, CH_3_Cl). **^1^H-NMR (300 MHz, DMSO-d_6_):** δ 8.10 (bs, 1H), 7.82 (bs, 1H), 7.32 (dt, *J* = 9.2, 5,3 Hz, 1H), 7.05 (dt, *J* = 9.2, 1.9 Hz, 1H), 6.92–6.75 (m, 4H), 4.66 (dq, *J* = 6.3, 5.1 Hz, 1H) 4.47 (dd, *J* = 11.5, 2.2 Hz, 1H), 4,32 (ddd, *J* = 7.3, 5.1, 2.2 Hz, 1H), 4.10 (dd, *J* = 11.5, 7.3 Hz, 1H), 1.36 (d, *J* = 6.3 Hz, 3H) ppm. **^13^C-NMR (75 MHz, DMSO-d_6_):** δ 161.7, 152.8 (dd, *J* = 242.2, 6.8 Hz), 149.4 (dd, *J* = 248.6, 8.6 Hz), 143.5, 143.3, 142.0 (dd, *J* = 11.0, 3.3 Hz), 122.0, 121.8, 118.7 (dd, *J* = 9.4, 2.2 Hz), 117.5, 117.4, 117.2 (dd, *J* = 24.8, 20.9 Hz), 111.6 (dd, *J* = 23.0, 4.0 Hz) 75.3, 75.1, 64.4, 16.0 ppm. Tr (HPLC): 6.01 min (Method A).

**2-(prop-1-en-2-yl)-1,4-benzodioxane (12):** Operating under a nitrogen atmosphere, potassium *tert*-butoxide (0.30 g, 2.7 mmol) was added to a solution of methyltriphenylphosphonium bromide (1.28 g, 3.0 mmol) in dry toluene (10 mL). After stirring for 30 min at RT, a solution of 2-acetyl-1,4-benzodioxane **1** (0.27 g, 1.5 mmol) in dry toluene (5 mL) was added dropwise and left stirring for 5 h at 80 °C. The reaction mixture was then concentrated under vacuum, diluted with ethyl acetate, washed twice with brine, dried over Na_2_SO_4_, filtered, and concentrated under vacuum to yield 0.18 g (69%) of **12** as a colorless oil residue. **^1^H-NMR (300 MHz, CDCl_3_):** δ 7.02–6.81 (m, 4H), 5.19 (s, 1H), 5.10 (s, 1H), 4.53 (dd, *J* = 8.1, 2.4 Hz, 1H), 4.30 (dd, *J* = 11.3, 2.4 Hz, 1H), 3.95 (dd, *J* = 11.3, 8.1 Hz, 1H), 1.85 (s, 3H) ppm.

**2-(1′-Hydroxypropyl)-1,4-benzodioxane l (13):** Operating under a nitrogen atmosphere, a solution of BH_3_ in THF 1M (3 mL) was added to a solution of 2-(prop-1-en-2-yl)-1,4-benzodioxane **12** (0.18 g, 1.0 mmol) in THF (5 mL). After stirring for 2 h, the reaction mixture was cooled down to 0 °C and sequentially added with H_2_O (1.8 mL), 10% aqueous NaOH (4 mL), and 30% H_2_O_2_ (3.6 mL). The reaction was kept stirring for a further 18 h, then diluted with H_2_O and extracted three times with ethyl acetate. The organic phase was dried over Na_2_SO_4_, filtered, and concentrated under vacuum to yield 0.13 g (68%) of **13** as a yellow oil. **^1^H-NMR (300 MHz, CDCl_3_):** δ 6.92–6.79 (m, 4H), 4.41–3.97 (m, 3H), 3.85–3.64 (m, 2H), 2.17–1.94 (m, 1H), 1.06 (d, *J* = 7.0 Hz, 3H).

**2-(1-mesiloxypropyl)-1,4-benzodioxane (14):** Methanesulfonyl chloride (89 μL, 0.54 mmol) was added dropwise to a solution of 2-(2,3-dihydro-1,4-benzodioxin-3-yl)propan-1-ol **13** (0.08 g, 0.4 mmol) and TEA (36 μL, 0.54 mmol) in DCM (1.6 mL) at 0 °C. The mixture was stirred at room temperature for 3 h, diluted with DCM, washed firstly with 10% aqueous NaHCO_3_, secondly with 10% aqueous HCl, and finally with brine, dried over Na_2_SO_4_, filtered, and concentrated under vacuum to yield 0.11 g (92%) of **14** as a yellow oil. **^1^H-NMR (300 MHz, CDCl_3_):** δ 7.00–6.62 (m, 4H), 4.51–4.38 (m, 1H), 4.36–4.16 (m, 3H), 4.09–3.95 (m, 2H), 3.03 (s, 3H), 1.16 and 1.11 (d, *J* = 7.0 Hz, 3H) ppm.

***Erythro* (FZ98) and *threo* (FZ97) 3-(2-(1,4-benzodioxan-2-yl)-1-propyloxy)-2,6-difluorobenzamide:** Operating under a nitrogen atmosphere, potassium carbonate (0.06 g, 0.44 mmol) was added to a solution of 2,6-difluoro-3-hydroxybenzamide (0.07 g, 0.4 mmol) in dry DMF (0.5 mL). After stirring at room temperature for 30 min, a solution of 1-mesiloxypropyl-1,4-benzodioxane **14** (0.11 g, 0.4 mmol) in DMF (1 mL) was added. The reaction mixture was stirred at 80 °C for 16 h, concentrated under vacuum, diluted with ethyl acetate (30 mL), washed with brine (3 × 20 mL), dried over Na_2_SO_4_, filtered, and concentrated to give 0.11 g of **3** and **4.** The obtained crude product was purified through flash chromatography on silica gel. Elution with 3/7 Cyclohexane/Ethyl Acetate gave 0.06 g (43%) as a mixture of stereoisomers and as a yellowish oil. The purification via preparative HPLC using a Water XBridgeTM C18 column (5 μm, 19 × 150 mm) and as eluent water + TFA 1% and ACN + TFA 1% 7:3 allowed for the separation and isolation of the *erythro* and *threo* derivatives.

***Erythro (*FZ98*)*: ^1^H-NMR (300 MHz, DMSO-d_6_):** δ 8.09 (bs, 1H), 7.81 (bs, 1H), 7.26 (dt, *J* = 9.3, 5.3 Hz, 1H), 7.05 (dt, *J* = 9.3, 1.9 Hz, 1H), 6.81 (m, 4H), 4.41 (dd, *J* = 11.4, 2.1 Hz, 1H), 4.20 (ddd, *J* = 8.4, 5.4, 2.1 Hz, 1H), 4.13 (dd, *J* = 9.7, 6.5 Hz, 2H), 4.04 (dd, *J* = 9.7, 6.0 Hz, 1H), 4.04 (dd, *J* = 11.4, 8.4 Hz, 1H), 1.09 (d, *J* = 7.0 Hz, 3H) ppm. **^13^C-NMR (75 MHz, DMSO-d_6_):** δ 161.7, 152.3 (dd, *J* = 241.1, 6.8 Hz), 148.3 (dd, *J* = 248.4, 8.5 Hz), 143.7, 143.6, 143.4 (dd, *J* = 11.0, 3.2 Hz), 121.7, 121.6, 117.5, 117.3, 117.04 (dd, *J* = 25.0, 20.4 Hz), 116.0 (dd, *J* = 9.6, 2.4 Hz), 111.4 (dd, *J* = 22.9, 4.0 Hz), 74.1, 71.1, 66.4, 34.5, 12.1 ppm. Tr (HPLC): 18.99 min (Method B).

***Threo (*FZ97*)*: ^1^H-NMR (300 MHz, DMSO-d_6_):** δ 8.08 (bs, 1H), 7.80 (bs, 1H), 7.22 (dt, *J* = 9.2, 5.3 Hz, 1H), 7.03 (dt, *J* = 9.2, 1.9 Hz, 1H), 6.80 (m, 4H), 4.38 (dd, *J* = 11.3, 1.8 Hz, 1H), 4.14 (m, 3H), 4.06 (dd, *J* = 10.4, 6.6 Hz, 1H), 2.20 (m, 1H), 1.10 (d, *J* = 7.0 Hz, 3H) ppm. **^13^C-NMR (75 MHz, DMSO-d_6_):** δ 161.7, 152.3 (dd, *J* = 241.1, 6.9 Hz), 148.3 (dd, *J* = 248.4, 8.6 Hz), 143.6, 143.5 (dd, *J* = 11.0, 3.0 Hz), 143.4, 121.8, 121.6, 117.6, 117.3, 117.04 (dd, *J* = 25.3, 20.6 Hz), 115.9 (dd, *J* = 9.6, 3.0 Hz), 111.4 (dd, *J* = 22.9, 4.5 Hz), 74.3, 70.8, 66.0, 34.6, 13.2 ppm. Tr (HPLC): 22.01 min (Method B).

**2-Methyl-3-butenoic acid (15):** 2.7 M *n*-BuLi (9.26 mL, 25 mmol) in heptane was added dropwise to a solution of diisopropylamine (3.85 mL, 27.5 mmol) in THF (15 mL) at −78 °C. The reaction mixture was stirred for 1 h at −78 °C, added with 2-Methyl-2-butenoic acid (1.00 g, 10 mmol) in THF (5 mL), stirred at RT for 2 h, poured into 20 mL of 10% aqueous HCl at 0 °C, and extracted with Ethyl acetate (2 × 20 mL). The organic phase was washed with brine (2 × 10 mL), dried over Na_2_SO_4_, filtered, and concentrated under vacuum, affording 0.85 g of **15** (85%) as a colorless oil. **^1^H-NMR (300 MHz, CDCl_3_):** δ 5.94 (ddd, *J* = 18.5, 10.2, 7.4 Hz, 1H), 5.17 (m, 2H), 3.18 (m, 1H), 1.31 (d, *J* = 7.1 Hz, 3H) ppm.

**3,4-Dibromo-2-methylbutanoic acid (16):** Bromine (0.43 mL, 8.49 mmol) was added dropwise to a solution of 2-methyl-3-butenoic acid **15** (0.85 g, 8.49 mmol) in DCM (10 mL) at 0 °C. The reaction mixture was stirred for 18 h at RT, quenched with Na_2_S_2_O_5_ (2 × 10 mL), washed with brine (10 mL), dried over Na_2_SO_4_, filtered, and concentrated under vacuum, affording 1.72 g **28** (78%) as a colorless oil, which was used as a mixture of 4 stereoisomers in the next step without further purification. **^1^H-NMR (300 MHz, CDCl_3_):** *first couple of stereoisomers*: δ 4.29 (m, 1H), 3.89 (dd, *J* = 10.6, 4.9 Hz, 1H), 3.85 (dd, *J* = 10.8, 5.0 Hz, 1H), 3.33 (m, 1H), 1.35 (d, *J* = 7.0 Hz, 3H) ppm. *Second couple of stereoisomers*: δ 4.76 (ddd, *J* = 11.3, 8.6, 3.2 Hz, 1H), 4.07 (dd, *J* = 10.7, 8.6 Hz, 1H), 3.62 (dd, *J* = 11.0, 10.7, 1H), 3.33 (m, 1H), 1.26 (d, *J* = 6.8 Hz, 3H) ppm.

**Methyl 3,4-dibromo-2-methylbutanoate (17):** Trimethyl orthoformate (1.37 mL, 12.56 mmol) and a catalytic amount of concentrated H_2_SO_4_ were added to a solution of 3,4-dibromo-2-methylbutanoic acid **16** (1.72 g, 6.62 mmol) in MeOH (20 mL) at 0 °C. The reaction mixture was stirred at reflux for 18 h, concentrated under vacuum, diluted with Ethyl acetate (20 mL), washed with phosphate buffer pH = 7 (10 mL), dried over Na_2_SO_4_, filtered, and concentrated under vacuum, affording 1.38 g (78%) of **17** as a yellowish oil, used as a mixture of *threo* and *erythro* isomers in the next step without further purification.

**^1^H-NMR (300 MHz, CDCl_3_):*** first stereoisomer*: δ 4.74 (ddd, *J* = 10.9, 6.5, 3.5 Hz, 1H), 4.01 (dd, *J* = 10.8, 6.5 Hz, 1H), 3.62 (dd, *J* = 10.9, 10.8 Hz, 1H), 3.25 (m, 1H), 1.24 (d, *J* = 6.8 Hz, 3H) ppm. *Second stereoisomer*: δ 4.31 (dt, *J* = 8.2, 4.9 Hz, 1H), 3.87 (dd, *J* = 10.5, 4.9 Hz, 1H), 3.83 (dd, *J* = 10.5, 4.9 Hz, 1H), 3.23 (m, 1H), 1.31 (d, *J* = 7.0 Hz, 3H) ppm.

***Erythro* and *threo* Methyl 2-(5,6,7,8-tetrahydro-1,4-naphthodioxan-2-yl)-2-methylacetate (18):** A solution of methyl 3,4-dibromo-2-methylbutanoate **17** (1.38 g, 5.04 mmol) in DMF (5 mL) was added to a solution of 5,6,7,8-tetrahydronaphthalen-2,3-diol (0.75 g, 4.58 mmol) and K_2_CO_3_ in DMF (10 mL) at RT. The reaction mixture was stirred at 70 °C for 18 h, concentrated under vacuum, diluted with Ethyl acetate (20 mL), washed with brine (5 × 10 mL), dried over Na_2_SO_4_, filtered, and concentrated under vacuum. Elution with 97/3 Cyclohexane/Ethyl acetate gave 0.25 g (15%) of ***erythro* 18** and 0.25 g (15%) of ***threo* 18**.

***Erythro* 18: ^1^H-NMR (300 MHz, CDCl_3_)**: δ 6.56 (s, 2H), 4.32 (ddd, *J* = 9.0, 6.9, 2.1 Hz, 1H), 4.24 (dd, *J* = 11.4, 2.1 Hz, 1H), 4.00 (dd, *J* = 11.4, 6.9 Hz, 1H), 3.73 (s, 3H), 2.86 (m, 1H), 2.65 (m, 4H), 1.73 (m, 4H), 1.24 (d, *J* = 7.1 Hz, 3H) ppm. ***Threo* 18: ^1^H-NMR (300 MHz, CDCl_3_)**: δ 6.59 (s, 1H), 6.57 (s, 1H), 4.32–4.24 (ddd, *J* = 9.0, 6.5, 2.2 Hz, 1H), 4.21 (dd, *J* = 11.4, 2.2 Hz, 1H), 3.98 (dd, *J* = 11.4, 6.5 Hz, 1H), 3.72 (s, 3H), 2.82 (m, 1H), 2.64 (m, 4H), 1.74 (m, 4H), 1.33 (d, *J* = 7.1 Hz, 3H) ppm.

***Erythro* 2-(5,6,7,8-tetrahydro-1,4-naphthodioxan-2-yl)propan-1-ol (19):** LiAlH_4_ (0.018 g, 0.47 mmol) was suspended in dry THF (2 mL) at 0 °C under a nitrogen atmosphere. The solution of *erythro* methyl 2-(5,6,7,8-tetrahydro-1,4-naphthodioxan-2-yl)-2-methylacetate **18** (0.13 g, 0.47 mmol) in THF (3 mL) was slowly added to the reaction. The mixture was then warmed to RT and stirred for 1 h; at completion, it was cooled to 0 °C and slowly quenched with Ethyl acetate (15 mL). The organic layer was washed with brine (3 × 10 mL), dried over Na_2_SO_4_, and concentrated under vacuum to give 0.12 g of **19** (quantitative yield) as a colorless oil. **^1^H-NMR (300 MHz, CDCl_3_):** δ 6.57 (s, 2H), 4.26 (dd, *J* = 9.3, 1.2 Hz, 1H), 4.01 (m, 2H), 3.75 (m, 2H), 2.64 (m, 4H), 2.01 (m, 1H), 1.71 (m, 4H), 1.03 (d, *J* = 7.0 Hz, 3H) ppm.

***Erythro* 2-(5,6,7,8-Tetrahydro-1,4-naphthodioxan-2-yl)prop-1-yl methanesulfonate (21):** Methanesulfonyl chloride (56 μL, 0.72 mmol) was added dropwise to a solution of *erythro* 2-(5,6,7,8-tetrahydro-1,4-naphthodioxan-2-yl)propan-1-ol **19** (0.12 g, 0.48 mmol) and TEA (100 μL, 0.72 mmol) in DCM (10 mL) at 0 °C. The reaction mixture was stirred at room temperature for 3 h, diluted with DCM (15 mL), washed firstly with 10% aqueous NaHCO_3_ (5 mL), secondly with 10% aqueous HCl (5 mL), and finally with 10% aqueous NaCl (10 mL), filtered, and concentrated under vacuum to give 0.12 g of **21** (75%) as a yellowish oil. **^1^H-NMR (300 MHz, CDCl_3_)**: δ 6.57 (s, 2H), 4.42 (dd, *J* = 9.7, 4.9 Hz, 1H), 4.32 (dd, *J* = 9.7, 3.8 Hz, 1H), 4.25 (m, 1H), 4.01 (m, 2H), 3.03 (s, 3H), 2.65 (m, 4H), 2.20 (m, 1H), 1.72 (m, 4H), 1.14 (d, *J* = 7.0 Hz, 3H) ppm.

***Erythro* 3-(2-(5,6,7,8-Tetrahydro-1,4-naphthodioxan-2-yl)prop-1-yloxy)-2,6-difluorobenzamide (FZ118):** A solution of 3-Hydroxy-2,6-difluorobenzamide (0.067 g, 0.386 mmol) in DMF (2 mL) was added to a solution of *erythro* 2-(5,6,7,8-Tetrahydro-1,4-naphthodioxan-2-yl)prop-1-yl methanesulfonate **21** (0.12 g, 0.37 mmol) and K_2_CO_3_ (0.056 g, 0.41 mmol) in DMF (2 mL) at RT. The reaction mixture was stirred at 70 °C for 2 h, concentrated under vacuum, diluted with Ethyl Acetate (20 mL), washed with 10% aqueous NaCl (5 × 10 mL), dried over Na_2_SO_4_, filtered, and concentrated under vacuum to give a brown residue. Elution with 6/4 Cyclohexane/Ethyl acetate on silica gel and subsequent treatment with IPE (1 mL) gave 0.025 g of **FZ118** (16%) as a white solid. Tr (HPLC): 18.0 min (Method C). M.p. = 159 °C. **^1^H-NMR (300 MHz, DMSO-d_6_):** δ 8.09 (s, 1H), 7.82 (s, 1H), 7.23 (dt, *J* = 9.3, 5.3 Hz, 1H), 7.04 (dt, *J* = 9.0, 1.7 Hz, 1H), 6.52 (s, 2H), 4.31 (dd, *J* = 11.2, 1.5 Hz, 1H), 4.08 (m, 4H), 2.56 (m, 4H), 2.23 (m, 1H), 1.64 (m, 4H), 1.09 (d, *J* = 7.0 Hz, 3H) ppm. **^13^C-NMR (75 MHz, DMSO-d_6_):** δ 161.7, 152.3 (dd, *J* = 240.0, 6.7 Hz), 148.3 (dd, *J* = 246.8, 8.2 Hz), 143.4 (dd, *J* = 11.2, 3.0 Hz), 141.4, 141.3, 129.7, 129.6, 117.0 (dd, *J* = 21.2, 20.1 Hz), 117.0, 116.97, 116.7, 116.66, 116.0 (dd, *J* = 9.4, 1.9 Hz), 111.4 (dd, *J* = 22.9, 3.4 Hz), 74.0, 71.1, 66.4, 34.4, 28.5, 23.3, 12.1 ppm.

***Threo* 2-(5,6,7,8-tetrahydro-1,4-naphthodioxan-2-yl)propan-1-ol (20):** LiAlH_4_ (0.035 g, 0.905 mmol) was suspended in dry THF (2 mL) at 0 °C under a nitrogen atmosphere. The solution of *threo* methyl 2-(5,6,7,8-tetrahydro-1,4-naphthodioxan-2-yl)-2-methylacetate **18** (0.25 g, 0.91 mmol) in THF (3 mL) was slowly added to the reaction. The mixture was then warmed to RT and stirred for 1 h; at completion, it was cooled to 0 °C and slowly quenched with Ethyl acetate (15 mL). The organic layer was washed with brine (3 × 10 mL), dried over Na_2_SO_4_, and concentrated under vacuum to give 0.19 g of **20** (85%) as a colorless oil. **^1^H-NMR (300 MHz, CDCl_3_):** δ 6.58 (s, 1H), 6.57 (s, 1H), 4.24 (dd, *J* = 10.8, 2.0 Hz, 1H), 4.17 (m, 1H), 4.03 (dd, *J* = 10.8, 8.0 Hz, 1H), 3.72 (m, 2H), 2.64 (m, 4H), 2.04 (m, 1H), 1.73 (m, 4H), 1.05 (d, *J* = 7.0 Hz, 3H) ppm.

***Threo* 2-(5,6,7,8-Tetrahydro-1,4-naphthodioxan-2-yl)prop-1-yl methanesulfonate (22):** Methanesulfonyl chloride (0.09 mL, 1.15 mmol) was added dropwise to a solution of *threo* 2-(5,6,7,8-tetrahydro-1,4-naphthodioxan-2-yl)propan-1-ol **20** (0.19 g, 0.76 mmol) and TEA (0.16 mL, 1.15 mmol) in DCM (10 mL) at 0 °C. The reaction mixture was stirred at room temperature for 3 h, diluted with DCM (15 mL), washed firstly with 10% aqueous NaHCO_3_ (5 mL), secondly with 10% aqueous HCl (5 mL) and finally with 10% aqueous NaCl (10 mL), filtered, and concentrated under vacuum to give 0.19 g of **22** (76%) as a yellowish oil. **^1^H-NMR (300 MHz, CDCl_3_):** δ 6.59 (s, 2H), 4.33 (dd, *J* = 9.0, 7.2 Hz, 1H), 4.26 (dd, *J* = 9.5, 4.3 Hz, 1H), 4.20 (m, 2H), 4.01 (dd, *J* = 11.7, 8.8 Hz, 1H), 3.03 (s, 3H), 2.66 (m, 4H), 2.30 (m, 1H), 1.75 (m, 4H), 1.10 (d, *J* = 7.0 Hz, 3H) ppm.

***Threo 3-(*2-(5,6,7,8-Tetrahydro-1,4-naphthodioxan-2-yl)prop-1-yloxy)-2,6-difluorobenzamide (FZ119):** A solution of 3-Hydroxy-2,6-difluorobenzamide (0.11 g, 0.61 mmol) in DMF (2 mL) was added to a solution of *threo* 2-(5,6,7,8-Tetrahydro-1,4-naphthodioxan-2-yl)prop-1-yl methanesulfonate **22** (0.19 g, 0.58 mmol) and K_2_CO_3_ (0.088 g, 0.64 mmol) in DMF (2 mL) at RT. The reaction mixture was stirred at 70 °C for 2 h, concentrated under vacuum, diluted with Ethyl Acetate (20 mL), washed with 10% aqueous NaCl (5 × 10 mL), dried over Na_2_SO_4_, filtered, and concentrated under vacuum to give a brown residue. Elution with 6/4 Cyclohexane/Ethyl acetate on silica gel and subsequent treatment with IPE (2 mL) gave 0.06 g of **10** (26%) as a white solid. Tr (HPLC): 17.78 min (Method C). M.p. = 170 °C. **^1^H-NMR (300 MHz, DMSO-d_6_):** δ 8.09 (s, 1H), 7.82 (s, 1H), 7.26 (dt, *J* = 9.3, 5.3 Hz, 1H), 7.06 (dt, *J* = 9.0, 1.8 Hz, 1H), 6.52 (s, 1H), 6.51 (s, 1H), 4.34 (dd, *J* = 11.4, 1.9 Hz, 1H), 4.13 (m, 2H), 3.99 (m, 2H), 2.56 (m, 4H), 2.28 (m, 1H), 1.64 (m, 4H), 1.07 (d, *J* = 6.9 Hz, 3H) ppm. **^13^C-NMR (75 MHz, DMSO-d_6_):** δ 161.7, 152.2 (dd, *J* = 240.0, 6.7 Hz), 148.3 (dd, *J* = 246.7, 8.2 Hz), 143.6 (dd, *J* = 10.8, 3.3 Hz), 141.3, 141.0, 129.8, 129.6, 117.0 (dd, *J* = 24.2, 21.3 Hz), 117.0, 116.99, 116.7, 116.66, 115.9 (dd, *J* = 9.0, 2.0 Hz), 111.4 (dd, *J* = 22.3, 2.8 Hz), 74.2, 70.9, 66.0, 34.4, 28.5, 23.3, 13.3 ppm.

### 4.2. Cells

Normal human lung fibroblasts (MRC-5) were grown in Dulbecco’s Modified Eagle’s medium (DMEM) supplemented with 10% heat-inactivated fetal calf serum (FCS), 100 U/mL penicillin, and 100 mg/mL streptomycin.

Gram-positive methicillin-sensitive *Staphylococcus aureus* (MSSA, ATCC 29213) was grown in Luria–Bertani broth (LB), as already described in our papers [11,12,13,14].

### 4.3. Antibacterial Activity

#### 4.3.1. MSSA and MDRSA Protocol

The detailed protocol is identical to what was previously reported [12,13,14]. Briefly, the antimicrobial testing of compounds followed the Clinical and Laboratory Standards Institute and European Committee for Antimicrobial Susceptibility Testing guidelines. A bacterial suspension equivalent to 0.5 McFarland turbidity standard was diluted in cation-adjusted Mueller Hinton broth to prepare a fresh bacterial culture of 10^5^ CFU/mL. Compounds dissolved in DMSO were combined with the bacteria and incubated for 20 h at 35 °C. MIC values were determined visually as the lowest compound dilution with no turbidity. MICs were determined against methicillin-sensitive and methicillin-resistant *S. aureus* strains (11.7 and 12.1). Every assay was performed in triplicate, and for each series of experiments, both positive (tetracycline) and negative (no compounds) controls were included.

#### 4.3.2. Antibacterial Activity against *B. subtilis*

The *B. subtilis* strain used was WM5126 (JH642 *amyE*::P_xyl_-*gfp-zapA*) [14]. For spot viability assays on plates, compounds were diluted from DMSO stocks into molten LB agar and mixed well before solidifying. To assess viability, cells were grown until the early log phase, and then 10-fold serial dilutions of the culture in LB were spotted onto LB agar plates containing the indicated final concentration of the compounds. The plates were incubated overnight at 30 °C and photographed. Microdilution assays to obtain MICs were performed essentially as described for *E. coli* [15].

For fluorescence microscopy, overnight cultures of WM5126 were grown at 33 °C (after 1:200 dilution overnight and addition of 0.1% xylose to induce expression of GFP-ZapA) for ~2 h until reaching an OD_600_ of ~0.2 and then treated with no drug or with compounds **FZ116** or **FZ117** diluted in water to a final concentration of 0.3 µg/mL (0.8 µM) from stock solutions of 30 mg/mL in DMSO. After the addition of the compounds, the cultures were grown for 90 min while shaking at 33 °C prior to spotting on a thin layer of 1% agarose in phosphate-buffered saline. Cells were imaged with an Olympus BX100 fitted with a 100 × Plan Apochromat objective (N.A. 1.4) and a GFP filter set (Chroma). Images were acquired with CellSens Dimension version 2 software (Evident Corporation, Tokyo, Japan).

### 4.4. Thiazolyl Blue Tetrazolium Bromide (MTT) Cytotoxicity Assay

Compounds showing promising antibacterial activity were serially diluted in DMEM and tested for cytotoxicity on MRC-5 cells using the MTT assay (Sigma, St Louis, MO, USA), following the same protocol we previously reported [11,12,13,14].

### 4.5. Computational Studies

#### 4.5.1. Ligand and Protein Preparation

Ligand Preparation: Conversion from SMILE to SD format was possible using the tool “structconvert” available in the Schrödinger module [26]. Preparation was carried out using the tool included in the Maestro package LigPrep [27,28]. Progressive levels were generated, including the generation of possible ionization states at physiological pH and the potential tautomers. Final energy minimizations were obtained using the OPLS4 force field. Default parameters for stereoisomers were set.

Protein Preparation: The protein was prepared for further computational studies using the Protein Preparation Wizard [29,30], a tool implemented in Maestro [28]. As part of the protocol, the structure of the protein was pre-processed by assigning bond orders, adding the corresponding hydrogens, and generating ligand protonation states at pH 7 ± 2 using Epik. Prime was used to induce adjustments in receptor structures. Optimization of the hydrogen bond network and calculation of the protonation states of the residues at pH 7 was achieved using PROPKA. Water molecules were avoided beyond everything that is not a protein residue in 3 A, followed by a final restraint minimization with the OPLS4 force field.

#### 4.5.2. Docking

Docking Study: Docking was performed employing the Glide module included in the Schrodinger software Release 2021-1 [31,32], and the PDB code of the used FtsZ is 5XDT [33]. The centre of the TXA707 ligand in the catalytic pocket was selected as the centroid of the grid. In the grid generation, a scaling factor of 1.0 in van der Waals radius scaling and a partial charge cutoff of 0.25 were used. The extra precision mode (XP) was used for the validation of the docking study; no constraints were applied during the process. Ligand parameters were selected by default. For the settings, the ligand sampling was flexible, and epik state penalties were added to the docking score. In the last step, a post-docking minimization was set as default.

Induced Fit Docking (IFD): The IFD technique allows for changes in the active site residues’ geometry using a complex structure [34,35,36,37]. These changes allow the protein to alter its binding site so that it more closely poses to the shape and binding mode of the new ligand. The IFD standard protocol was based on Glide and the Refinement module in Prime for accurate prediction of ligand binding modes. Parameters for Initial Glide docking were set by default using a maximum of 20 poses per ligand retained. Prime was able to predict the protein structure by using the pose of the corresponding ligand and through a rearrangement of nearby side chains of the active site and a minimization of the whole FtsZ energy. Finally, each drug was redocked with Glide XP mode into its corresponding low. All residues within 5.0 Å of the ligand poses were refined. The rest of the parameters were set as default. In order to optimize the active site of the protein considering the new set of 2,6-difluorobenzamide derivatives, an IFD was performed using the crystallized ligand as a reference compound.

#### 4.5.3. Validations

RMSD calculation: RMSD values were obtained by employing the Superposition module included in the Schrodinger software [28].

Physicochemical properties. QikProp is a module implemented in Maestro that accurately predicts important molecular descriptors and pharmaceutical properties for organic compounds. For settings, fast mode was selected, and five drug molecules were used to identify similarity. The #stars parameter, which compares property or descriptor values that are beyond the 95% range of comparable values for recognized medications, was also considered, and it was shown to be the best option for all of the compounds. In detail, the parameters evaluated for #stars were MW, dipole, IP, EA, SASA, FOSA, FISA, PISA, WPSA, PSA, volume, #rotor, donorHB, accptHB, glob, QPpolrz, QplogPsC16, QPlogPoct, QPlogPw, QPlogPo/w, logS, QPLogKhsa, QPlogBB, and #metabol [38,39].

## Data Availability

The data presented in this study are available upon request from the corresponding author.

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
