# Peer review of "Benzodioxane–Benzamides as FtsZ Inhibitors: Effects of Linker’s Functionalization on Gram-Positive Antimicrobial Activity"

_antibiotics, 2023, doi:10.3390/antibiotics12121712_

Round 1
Reviewer 1 Report
Comments and Suggestions for Authors
The authors synthesized 13 benzodioxane-benzamide derivatives to be tested as FtsZ inhibitors on MSSA and MRSA, and the effect of the compounds on FtsZ assembly on B. subtilis. The authors provided a comprehensive overview of the chemical synthesis of the FZ compounds, computational studies and conducted some biological assays to determine the antimicrobial activity of the compounds.
The chemical synthesis and computational studies of the compounds were in-depth and there are no further comments.
An area that was slightly ambiguous was whether the biological activity of the compounds was tested separately, or in combination? For example, line 127 implied that the compounds were used in combination, but the results in Table 1 suggested otherwise. If the authors were trying to group them based on their stereoisomers, perhaps, this could also be made clear.
The authors also tested the antimicrobial activity of the FZ compounds on the FtsZ of B. subtilis. I felt an area that was lacking was the effect of the compounds on the GTPase and polymerization of the FtsZ protein. They are crucial information to determine if the compounds indeed target FtsZ.
Off-target effects have been displayed in some FtsZ benzamide derivatives by means of inhibiting mammalian tubulin. I wonder if the authors could select some of the more potent FZ compounds to establish if this might be the case. In particular, FZ116 and FZ117 since they are promising. Additionally, the authors described having performed cytotoxicity analysis with ‘low to no cytotoxicity towards human MRC-5 cells’. I suggest having at least several more cell lines (such as liver or kidney) to rule out the potential toxic effects of the compounds, not just a single human lung fibroblast cell line.
There have also been merits with the benzamide derivatives shown to reverse resistance in beta-lactam antibiotics. Considering MDRSA 12.1 and 11.7 are resistant to a range of antimicrobial agents, it will be interesting to determine if FZ116 and FZ117 can display reversal of resistance to beta-lactam antibiotics that are used clinically.
The relevance of this research is of current interest where the scientific and clinical context is significant. I recommend it to be published in the Antibiotics after the revision from above.
Reviewer 2 Report
Comments and Suggestions for Authors
The current manuscript can be accepted for publication on condition that the authors respond to the following comments and inquiries. Upon receiving the authors’ response, the manuscript can be accepted for publication.
1. In scheme 1 typo errors were observed. The authors should change Reagents and solvents as follows: a) NaBH4, MeOH, 0 °C; (b) methanesulfonyl chloride (MsCl), triethylamine (TEA), dichloromethane (DCM), RT; (c) 2,6-difluoro-3-hydroxybenzamide, K2CO3, dimethylformamide (DMF), 80 °C.
2. The authors should provide the PDB ID for the crystal structure of FtsZ used for the docking study.
3. In Table 2, please provide an additional column for the molecular weight property.
Also, in Table 2, please explain how compounds FZ118RR, FZ118SS, FZ119RS, and FZ119SR are predicted to have low human oral absorption (1) but 100% human oral absorption.
4. Would the authors please provide a graph correlating the predicted binding affinity vs. experimental activity data?
5. Please provide IFD scores for all docked compounds.
6. For section 2.4.2, please report the RMSD for the docked pose of TXA707. Also, please report the key interactions made by the co-crystalized ligand or provide a figure of the co-crystalized ligand within the binding domain.
7. To differentiate enantiomers, the experimental section should include the specific rotation of all the compounds.
8. The authors should provide the mass data of all the synthetic derivatives.
9. In the supporting information, the solvent used for measuring the NMR should be changed to DMSO-d6 and also provide peak picking values in 13C NMR spectrum.
10. The authors provide HPLC profiles of compounds FZ97, 98, 104, 105, 112, 113, 116, 117, 118 and 119 in supplementary information.
Reviewer 3 Report
Comments and Suggestions for Authors
Reviewer’s Comments to the Author
The current research manuscript describes Benzodioxane-benzamides as FtsZ inhibitors: effects of linker’s functionalization on Gram-positive antimicrobial activity. In addition to the OH group survey, the authors explored the effect of substituent on the ethylenic linker between the two main moieties and evaluated antimicrobial activity and pharmacokinetic properties. In the reviewer’s opinion, the study adds a new finding to the existing literature to display interesting and informative aspects. The authors have also made decent efforts to come up with this finding. Herein, I would like to recommend a publication of the current version. To this end, there is one suggestion related to the manuscript and authors should take into account these suggestions before further submission.
1) Scheme 1 needs correction. Reviewer feels that the order of reagents is not appropriate. For example, describe “d”.
2) The author introduced, CH3 and compared it with the OH group. Reviewer feels that it would have been much better if CH3 was compared with CF3. This would have given a better comparison related to what is mentioned in lines 323-329.
Reviewer 4 Report
Comments and Suggestions for Authors
Review of :' Benzodioxane-benzamides as FtsZ inhibitors:" by Suigo et al.
The authors investigate the effects of various substitutions in the linker region of benzodioxane-benzamides and have characterized these new compounds with respect to their antimicrobial effects against S. aureus and B. subtilis. The effects of a subset of the compounds on FtsZ activities were examined in B subtilis and the predicted physicochemical properties of the new compounds were calculated. Finally, modeling examined the interaction of some of the more promising compounds with FtsZ.
This is an interesting and straightforward paper that describes advances in our understanding of a new class of antimicrobials.
The following are small points only:
1: How conserved among bacteria are the FtsZ residues (featured in Figure 7) that are proposed to be critical for drug-FtsZ interaction?
2: line 227; the 'white arrows' are yellow in Figure 5 on my pdf.
3:
Line 41; replace 'wide spreading' with 'dissemination' or 'emergence'.
Line 138; 'although' is mis-spelled.
Round 2
Reviewer 2 Report
Comments and Suggestions for Authors
The authors have responded to all my comments and suggestions. As such, the manuscript can be acceptable for publication in its current status.